# Text Emotion Recognition Based on XLNet-BiGRU-Att

**Tian Han** [1,2][iD], **Zhu Zhang** [1,2,*][iD], **Mingyuan Ren** [1][iD], **Changchun Dong** [1], **Xiaolin Jiang** [1] **and Quansheng Zhuang** [2]

1 Jinhua Advanced Research Institute, Jinhua 321013, China; htopen@foxmail.com (T.H.);
  rmy2000@126.com (M.R.); hitdongcc@163.com (C.D.); jlynner@163.com (X.J.)
2 School of Measurement and Communication Engineering, Harbin University of Science and Technology,
  Harbin 150080, China; izqshust@tom.com
* Correspondence: zhuzhang.zz@foxmail.com

**Abstract:** Text emotion recognition (TER) is an important natural language processing (NLP) task which is widely used in human–computer interaction, public opinion analysis, mental health analysis, and social network analysis. In this paper, a deep learning model based on XLNet with bidirectional recurrent unit and attention mechanism (XLNet-BiGRU-Att) is proposed in order to improve TER performance. XLNet is used to build bidirectional language models which can learn contextual information simultaneously, while the bidirectional gated recurrent unit (BiGRU) helps to extract more effective features which can pay attention to current and previous states using hidden layers and the attention mechanism (Att) provides different weights to enhance the 'attention' paid to important information, thereby improving the quality of word vectors and the accuracy of sentiment analysis model judgments. The proposed model composed of XLNet, BiGRU, and Att improves performance on the whole TER task. Experiments on the Interactive Emotional Dyadic Motion Capture (IEMOCAP) database and the Chinese Academy of Sciences Institute of Automation (CASIA) dataset were carried out to compare XLNet-BiGRU-Att, XLNet, BERT, and BERT-BiLSTM, and the results show that the model proposed in this paper has superior performance compared to the others.

**Keywords:** text emotion recognition; XLNET; BiGRU; attention; BERT

## 1. Introduction

The area of text emotion recognition has garnered significant interest in recent years, largely due to the proliferation of digital communication devices. Text emotion recognition involves analyzing the emotions expressed within text to gain a more comprehensive understanding of its contents, including its underlying sentiment, purpose, and expression [1]. This technology has been widely adopted across different industries, including business, education, and communication [2].

The underlying principle behind detecting emotional content in text is rooted in natural language processing, an expanding research domain that has gained tremendous momentum in the wake of a burgeoning volume of online comments. The core operation in recognizing emotional content in text involves a systematic text preprocessing phase that entails segmenting words, annotating parts of speech, and analyzing syntactic and semantic structures, among others [3]. The subsequent stage is text feature extraction, in which quantized word vectors derived from the text and fed into the classifier to categorize emotions [4,5]. These extracted features are represented in a format amenable to recognition by computer algorithms. Currently, numerous widely employed algorithms exist for extracting statistical features [6]. One-hot encoding is an encoding technique that yields a binary representation of classified values, and has proven moderately effective [7]. Another commonly used text representation technique is the Term Frequency–Inverse Document Frequency (TF-IDF) method, which explores word frequency to create a simple and rapid representation of text [8–10]. In summary, this technology serves as a means of mitigating noise by assigning weights to different elements. The algorithm involves

calculating the frequency of words or phrases in a particular text and the frequency of those same elements across the entire corpus. While the idea behind the algorithm is relatively straightforward, particular attention must be paid to selecting effective features, determining feature weights, and designing classification algorithms. Recent proposals in the literature include a novel network structure comprising a Bidirectional Long Short Term Memory (BiLSTM) model and hierarchical attention mechanism [11,12]. Within this structure, data are encoded and transmitted to the BiLSTM using a single key, then the output is stratified. Finally, the processed features are classified using a softmax classifier. Experimental results demonstrate the structure's high accuracy in text classification.

The Bidirectional Encoder Representations from Transformers (BERT) pre-trained model, has been hailed as a breakthrough in Natural Language Processing (NLP) due to its superior level of language comprehension [13]. However, there are certain limitations associated with the BERT model, such as its fixed input length size and word piece embedding issues, as well as computational complexities [14,15]. To address these concerns, several new pre-trained models such as Generalized Auto-regression Pre-training for Language Understanding (XLNet) [16], Robustly optimized BERT pre-training Approach (RoBERTa) [17], and DistilBERT [18] have been proposed. The effectiveness of BERT and its variants in the field of NLP has led researchers to shift their focus towards exploring text emotion recognition using this methodology [19,20]. This article presents an enhanced model of XLNET utilizing this approach for the purpose of accurately recognizing emotions within text.

In summary, to improve recognition accuracy based on existing text emotion recognition technologies, we need to optimize several problems. First of all, when processing information, we need to integrate and understand contextual information. secondly, we need to optimize the model to reduce dependence on long text. Thirdly, we need to enhance the representation and extract more effective features of text. Finally, it is better to reduce the complexity of the model. Based on the above objectives, we proposed a model composed of XLNet, BiGRU and Att. XLNet can process short sentences and obtain relatively rich semantic information by using context information to achieve effective representation of text. BiGRU can also combine the context bidirectional semantic information to effectively extract semantic features, while the parameter quantity is significantly reduced compared to LSTM. The attention mechanism can further allocate the importance of each feature through weights, achieving effective emotional classification output. In this paper, the proposed model for TER is presented in detail, comparative experiment is performed and the result is discussed.

## 2. Materials and Methods

### 2.1. Dataset

In this research, text data from two emotion datasets, namely, CASIA and IEMOCAP, were utilized to verify the model proposed.

The CASIA Chinese emotional corpus was initially curated by the esteemed Institute of Automation at the Chinese Academy of Sciences. The repository comprises of recordings by four proficient speakers and encompasses six distinct emotional states (anger, happiness, sadness, fear, surprise, and neutrality), totalling an impressive 9600 pronunciations. A noteworthy aspect of the corpus is that while it contains 300 identical texts, the different emotional renderings of each exhibit diverse acoustic and rhythmic performances when subjected to precise comparative analysis. In addition, the corpus features 100 distinct texts which the literal meaning suggests as having inherent emotional leanings, facilitating more consistent and accurate emotion depiction. For our experiment, we handpicked the identical texts in corpus for use a experimental data, with the first 200 readings assigned to the training set and the residual 100 allocated for testing. Furthermore, the experiment involved partitioning of the dataset [21].

The Interactive Emotional Binary Motion Capture Database (IEMOCAP) was collected by the Speech Analysis and Interpretation Laboratory (SAIL) of the University of Southern

California (USC). This database is an invaluable resource for research and modeling in the field of multimodal and expressive human communication; it contains over twelve hours of data recorded from ten actors engaged in binary conversation scenes along with detailed motion capture information that identifies the subjects' facial expressions and hand movements. Additionally, the database includes interactive settings stimulating specific emotions such as happiness, anger, sadness, and depression in emotionally scripted and naturally occurring conversation scenarios [22].

*2.2. Text Emotion Recognition Model Based on XLNet*

This paper proposes a novel approach to enhance the accuracy of sentiment analysis through the development of a model based on XLNet with bidirectional recurrent unit and attention mechanism (XLNet-BiGRU-Att). The proposed methodology addresses the challenges associated with vectorization of short text sentiment analysis. Unlike traditional language models that only consider unidirectional information, the proposed XLNet model is a two-way modeling language model that simultaneously captures information in context both above and below, making the resulting word vectors more semantically rich. In addition, the BiGRU-Attention network layer effectively filters important information within limited text space and assigns different weights to filtered word vectors in order to enhance their "attention power". These measures collectively improve the performance of the sentiment analysis model. The architecture of XLNet-BiGRU-Att is shown in Figure 1, which is mainly composed of thr input layer, XLNet-BIGRU-Att model layer, and output layer. In the input layer, every word in the input sentence is transferred into a word vector by the embedding function, where the raw word vectors $[x_1, x_2, \cdots, x_t]$ represent the input of XLNet. In XLNet, the raw word vectors are processed by two-stream self-attention and the vectors are calculated concurrently in two channels: the content stream $h_T^{(2)}$, and the query stream $g_T^{(2)}$. As the query stream contains the position information, it is used as the output of XLNet during the prediction process. The BiGRU layer is used for extraction of deep emotional features; it contains both forward and backward GRUs, as shown in Figure 1, where $h$ is the hidden state. Through BiGRU, word vectors can be used to more fully learn the relationships between contexts and perform semantic encoding. The attention layer assigns corresponding probability weights to different word vectors in order to further extract text features and highlight the key information of the text. In the end, the output layer is a fully connected layer and a softmax function is used to provide the emotional classification result.

XLNet is a state-of-the-art model for pre-training in semantic understanding. It builds upon previous models such as mask language models and autoregressive (AR) language models, and addresses specific challenges in the pre-training stage of BERT. One key advantage of XLNet is its ability to handle inconsistencies between the mask flag and the fine-tuning process. Additionally, it effectively resolves the dependency problem between masked words. This is achieved through a unique approach that reconstructs the input text in a permutation and composition manner. Unlike BERT, XLNet applies this approach during the fine-tuning stage, using the Transformer attention mask matrix and double-flow attention mechanism to achieve different combinations and permutations. This allows for the integration of contextual features into the model training process.

XLNet is based on the autoregression (AR) language model. XLnet uses the idea of random sorting to solve the problem of AR models being unable to introduce two-directional text information. The random sorting process simply sorts the sequence number of each position randomly. After sorting, the words prior to each individual word can be used to predict its probability. Here, the previous words may be drawn from either the words prior to the current word in the original sentence or from the words after it, which is equivalent to using the bidirectional sequence information.

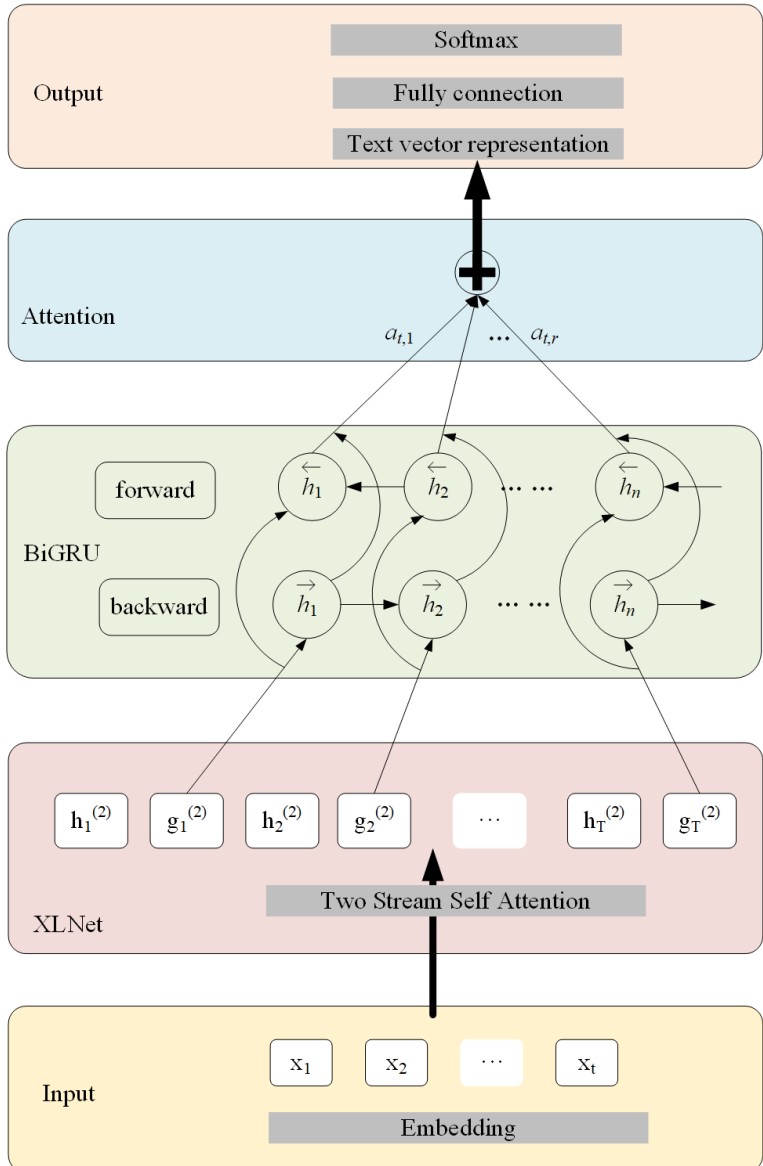

**Figure 1.** The XLNet-BIGRU-Att architecture.

First, the Permutation Language Model (PLM) is constructed, which disrupts the order of the original word vectors. Assuming that the original sequence is $x(x_1, x_2, x_3, x_4)$, then its arrangement is $(1, 2, 3, 4)$ and the word order is fully arranged as $(1, 3, 4, 2)$, $(2, 4, 3, 1)$, $(3, 2, 4, 1)$, etc. Figure 2 presents an example of the prediction of $x_3$ based on the different orders produced by sequential factorization.

In Figure 2, $x_i$ represents the input word vectors, $mem^{(0)}$ is the hidden state of the input layer, $mem^{(1)}$ is the hidden state of the first layer, $h_i^{(1)}$ is the predicted result of the $i$th word in the first layer, $h_i^{(2)}$ is the predicted result of the $i$th word in the second layer, and $x_3$ is the output predicted result, which can be understood as a word vector encoding. As shown in Figure 2a, when the factorization order is $(3, 2, 4, 1)$, the prediction of $x_3$ cannot involve attention to any other words, and can only be predicted based on the previous hidden state $mem$. As shown in Figure 2b, when the factorization order is $(2, 4, 3, 1)$, the semantics of $x_3$ can be predicted according to the meaning of $x_2$ and $x_4$. Unlike the AR language model, this method captures the contextual semantic information of $x_4$, which achieves the purpose of obtaining contextual semantics.

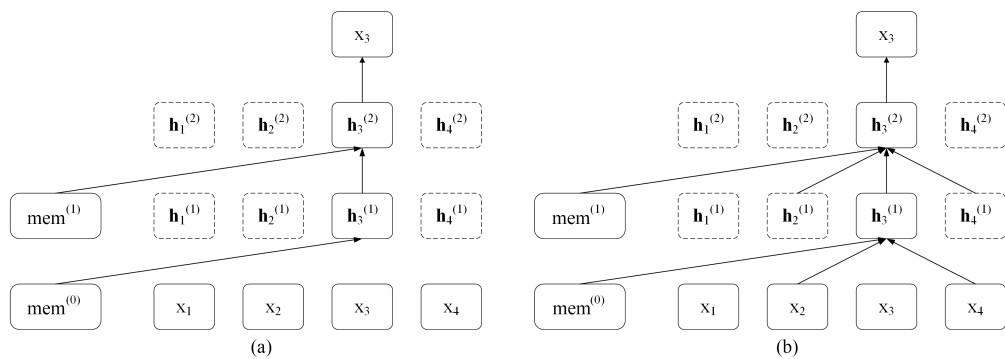

**Figure 2.** Different orders produced by sequential factorization: (**a**) factorization order 3, 2, 4, 1 and (**b**) factorization order 2, 4, 3, 1.

Second, all the factorization orders should be sampled. For a given length $T$ of text sequence $x$, $T!$ results can be obtained by fully sorting the sequence. However, when the text sequence is too long, the complexity of the algorithm increases. The random sorting language model needs to randomly sample $T!$ results to remove useless sequences. Random sampling of all the factorization orders of the text sequence is realized by Equation (1):

$$\max_{\theta} E_{z \sim Z_T} \left[ \sum_{t=1}^{T} \lg p_\theta \left( x_{Z_t} | x_{z<t} \right) \right] \qquad (1)$$

In Equation (1), $Z_T$ is the set of sequences with length $T$ composed of sequences arranged from the original sequence, $z$ is the sequence sampled from $Z_T$, $z_t$ is the value at the position of $t$ in sequence $z$, $z < t$ represents the $t$th element and the preceding $(t-1)$ elements in sequence $z$, and $E_{z \sim Z_T}$ is the expectation of the sampling result.

Finally, the later words in the sequence are predicted; for example, the original language sequence is arranged in the order $(1, 2, 3, 4)$ and the language sequence after it is fully arranged as $(2, 3, 4, 1)$ and $(1, 4, 2, 3)$. When the predicted position is $x_3$, the word order $(2, 3, 4, 1)$ can only be used to obtain the semantic information of $x_2$, while when the word order is $(1, 4, 2, 3)$, the semantic information of $x_1$, $x_4$, and $x_2$ can be obtained. Therefore, the model prefers that the predicted words be located at the end of the sentence, as this can provide better contact with the context semantic information.

### 2.2.1. Attention Mask in XLNet

The principle of the attention mask mechanism is to cover the part to be predicted inside the transformer to ensure that it does not play any role in the prediction process. For example, the original language sequence arrangement is $(1, 2, 3, 4)$ and the sampling sequence order is $(2, 4, 3, 1)$. The first line of the mask is used to predict $x_1$, the second line of the mask is used to predict $x_2$, and so on. The white dot means that the word in this position is masked, while the black dot means that the information of the word in this position can be contacted. As shown in Figure 3, when the semantic information of $x_3$ is predicted, the context of $x_2$ and $x_4$ can be used; thus, the positions of $x_2$ and $x_4$ in the third line of the mask are shaded. Because the sequence order is $(2, 4, 3, 1)$ and $x_2$ is first in the sequence, there is no reference information available for $x_2$ and the second line is fully masked. Following this theory, the mask matrix of the sequence $(2, 4, 3, 1)$ is shown in Figure 3. Black circle indicates the available data, and white circle means the data is masked.

The attention mask approach is used in PLM to realize random sorting and predict word vectors in different sequence orders. On the one hand, PLM solves the shortcoming of AR language models with respect to contacting the contextual semantics; on the other hand, PLM uses a mask matrix to replace the [mask] tag in the BERT model while maintaining the dependency between multiple predictors. For example, for a given sequence $(x_1, x_2, x_3, x_4,$

$x_5$, $x_6$, $x_7$), for XLNet, to predict $x_5$ and $x_3$ the random sampling sequence can be selected as $(1, 4, 2, 3, 5, 7, 6)$. Then, the probability of $x_5$ and $x_3$ is $p(x_5|x_1, x_2, x_3, x_4) + p(x_3|x_1, x_2, x_4)$. When the model predicts the semantics of the word $x_3$, the information of $x_5$ does not affect the prediction of $x_3$. When the semantic information of $x_5$ is predicted, however, the model can use the semantic information of $x_3$, thereby solving the problem that in other models all of the semantic information is independent and cannot be connected. This makes PLM an important theoretical foundation for the excellent performance of XLNet.

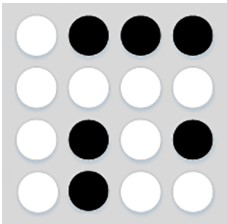

**Figure 3.** Mask matrix of the sequence (2,4,3,1).

### 2.2.2. Two-Stream Self-Attention Mechanism

While PLM can allow XLNet to relate to the context semantics, the original AR language model does not consider the location information of the words used in prediction. For example, given a sequence $(x_1, x_2, x_3, x_4)$, one of the permutations is $(2, 4, 3, 1)$. When the predicted statement is $x_3$, its probability is $p(x_3|x_2, x_4)$. The second arrangement order is $(2, 4, 1, 3)$. When the predicted word is $x_1$, its probability is $p(x_1|x_2, x_4)$. At this time, the semantic equivalent probability of the two is predicted; in fact, however, the semantic content of $x_1$ and $x_3$ is largely different. The main reason for this problem is that AR language models predict words based on the content before the predicted word, and as such do not need to consider the position information of the words in the word order. On the other hand, the random sorting language mechanism requires the word order to be rearranged completely. After the position information of the words is disrupted, the model cannot determine the position information of the predicted words in the original sequence. Therefore, we proposed solving this problem using the two-stream self-attention mechanism.

The two-stream self-attention mechanism can solve the problem of the position of the target prediction word being ambiguous due to the random disordering of the word order by the random sorting language model. Two-stream attention consists of the content stream and query stream. The objective function of the traditional AR language model for a sequence with length $T$ is shown in Equation (2):

$$P_\theta(X_{zt} = \mathrm{x}|x_{z<t}) = \frac{\exp\left(e(x)^\mathrm{T} h_\theta(x_{z<t})\right)}{\sum_{x'} \exp\left(e(x)^\mathrm{T} h_\theta(x_{z<t})\right)} \tag{2}$$

In the equation, $z$ is the sequence obtained by full array random sampling from a sequence $x$ with length $T$, $zt$ represents the sequence number of the position of $t$ in the sampling sequence, $x$ is the word to be predicted, and $e(x)$ is the embedding of $x$. The content hidden state $h_\theta(x_{z<t})$ encodes the content of $x$ and additionally encodes the above information, while not containing any location information, while $p_\theta(X_{z_t}|x_{z<t})$ demonstrates that for the prediction of words in position $t$ in the sorting sequence, the probability is calculated from the words corresponding to the sequence number before the position of $t$. Because PLM disrupts the sequence order, it is necessary to "explicitly" add the location information of the word to be predicted into the original sequence, meaning that Equation (2) is updated as Equation (3):

$$P_\theta(X_{zt} = \mathrm{x}|x_{z<t}) = \frac{\exp\left(e(x)^\mathrm{T} g_\theta(x_{z<t}, z_t)\right)}{\sum_{x'} \exp\left(e(x)^\mathrm{T} g_\theta(x_{z<t}, z_t)\right)} \tag{3}$$

In the equation, $z$ represents the query implicit state, including the words before $t$ position and the position information of the word $x$ to be predicted; it only encodes the context and location information of the prediction word $x$, and does not encode the content information of $x$. The updating processes of the content hidden state $h_\theta(x_{z<t})$ and query hidden state $g(x_{z<t}, z_t)$ are shown in Equations (4) and (5), respectively, where $m$ represents the number of layers in the network layer. Usually, the query hidden state $g(0)$ is initialized as a variable $w$ in layer 0 and the content hidden state $h(0)$ is initialized as the embedding of the word, that is, $e(x)$. The first layer of data is calculated according to layer 0 and then calculated layer by layer, with $Q$, $K$, and $V$ being the results of linear transformation of the input data according to different weights.

$$h_{z_t}^{(m)} \leftarrow Attention\left(Q = h_{z_t}^{(m-1)}, KV = h_{z \leq t}^{(m-1)}; \theta\right) \tag{4}$$

$$g_{z_t}^{(m)} \leftarrow Attention\left(Q = g_{z_t}^{(m-1)}, KV = h_{z \leq t}^{(m-1)}; \theta\right) \tag{5}$$

The purpose of the two-stream self-attention mechanism is to obtain the location information of $x_t$ without obtaining the content information when predicting a word $x_t$. For words other than $x_t$, the location information and content information should be provided. For example, if the given sequence is $(x_1, x_2, x_3, x_4)$ and the sampling sequence is assumed to be $(3, 2, 4, 1)$, the probability of the word with sequence number 1 can be predicted and then its probability can be calculated according to $x_3$, $x_2$, and $x_4$. The working principle used for calculating the probability of $x_1$ using the content stream and query stream is shown in Figure 4.

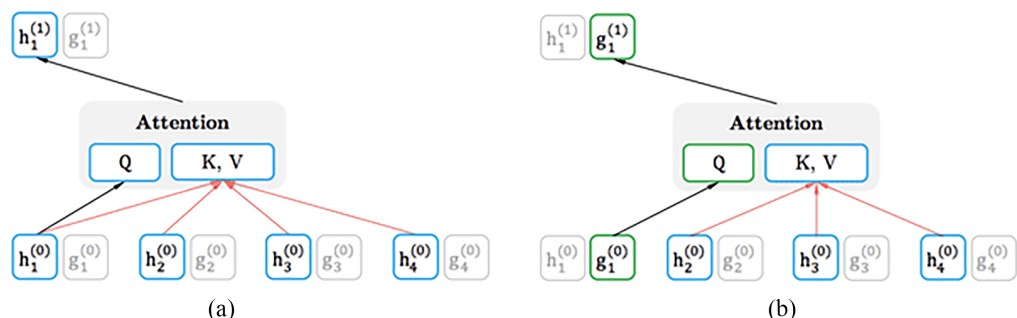

(a)　　　　　　　　　　　　　　　　(b)

**Figure 4.** Schematic diagram of content stream and query stream: (**a**) content stream and (**b**) query stream.

As shown in Figure 4a, the content stream encodes both the context information, and the self-information of the predicted word. As shown in Figure 4b, the query stream encodes the location information of the predicted word and other content information with the exception of the self-information of the predicted word.

Figure 5 shows the principle of two-stream self-attention when the sampling order of the sequence $(x_1, x_2, x_3, x_4)$ is $(3, 2, 4, 1)$. From the bottom to the top of Figure 5, starting from layer 0, the content stream $h$ and query stream $g$ are initialized by $e(x)$ and $w$, respectively. The first layer output $h(1)$ and $g(1)$ are respectively calculated through the content mask and query mask, then the second layer output is calculated in the same way. It can be observed from the mask matrix on the right that calculating the content stream is a standard transformer calculation process. When predicting the semantics of $x_1$, the semantic information of all words can be used, and when predicting the semantics of the word $x_4$, the semantic information of $x_2$, $x_3$, and $x_4$ can been used. In the content stream, the semantic information of the word itself can participate in predicting itself. When calculating the query stream, the attention mask functions somewhat differently. When predicting the semantic content of word $x_1$, its own content is masked, and the word is predicted only through the semantic content of $x_2$, $x_3$, and $x_4$. Similarly, when predicting $x_2$, only use

the semantic information of $x_3$ can be used. According to the above explanation, it can be deduced that in the query stream the information of the predicted word itself is not used for its own prediction. The difference between the content stream and query stream is whether the self-semantic information of the predicate itself.

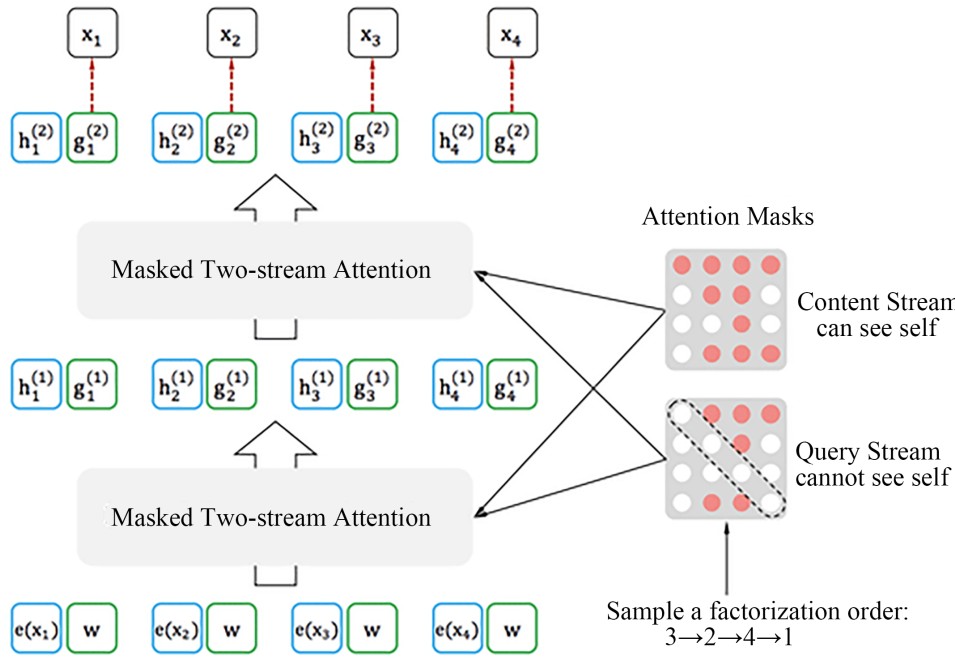

**Figure 5.** Schematic diagram of the two-stream self-attention mechanism.

### 2.3. Bi-Directional Gated Recurrent Unit

A gated recurrent unit (GRU) is a type of lightweight recurrent neural network that differs from other neural networks due to its internal gate structure. This unique structure enables the network to determine which data are relevant and which data can be discarded based on their relationship. This structure facilitates effective data transmission within the network while effectively controlling redundant information. As a result, the GRU partially addresses the issue of long-term dependence in neural networks. As shown in Figure 6, the structure of the GRU consists of three main components: the reset gate ($r_t$), update gate ($z_t$), and hidden state ($h_t$); these work together to extract temporal information and obtain long-term dependencies. In addition, $h_{t-1}$ is the hidden state of the prior time point and $\tilde{h}_t$ is the candidate hidden state. The individual functions of these gates and hidden states are described in further detail below, while the structure of the GRU is displayed in Figure 6.

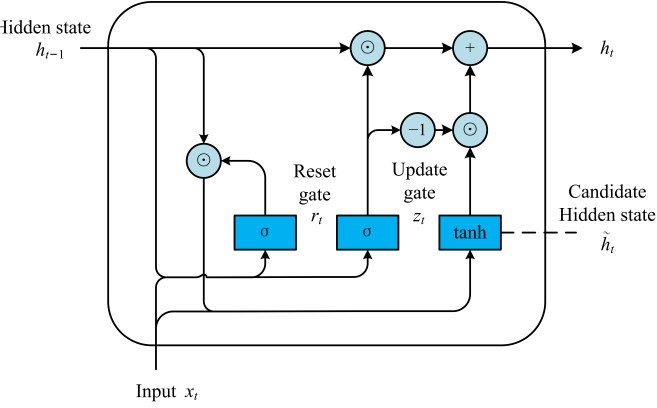

**Figure 6.** Structural diagram of a gated recurrent unit.

The input for the reset and update gates is obtained through the fully connected layer that operates on the current input $x_t$ and the hidden state of the previous time point $h_{t-1}$. The reset gate $r_t$ is calculated by Equation (6), and $\sigma$ is a Sigmoid function which is used as an activation function. Thus, the result of the reset gate is a vector with a value between 0 and 1 which is similar to the gate in circuit. This value is then used to determine the relevancy of the hidden state of the previous time point. The candidate hidden state $\widetilde{h}_t$ is determined by the input $x_t$, reset gate $r_t$, and prior hidden state $h_{t-1}$, as shown in Equation (7); the prior hidden state contains the information of both the current time and previous time, and is an important component of the current hidden state, although it is not the real current hidden state.

$$r_t = \sigma(x_t w_{xr} + h_{t-1} w_{hr} + b_r) \tag{6}$$

In the above equation, $t$ is the current time point, $t-1$ is the last time point, $x_t$ is the input vector of the current time, $h_{t-1}$ is the hidden state of the previous time point, $w_{hr}$ and $w_{xr}$ make up the weight matrix of the reset gate, and $b_r$ is the bias of the reset gate.

$$\widetilde{h}_t = \tanh(x_t w_{xh} + (r_t \odot h_{t-1}) w_{hh} + b_h) \tag{7}$$

Here, $w_x h$ and $w_h h$ make up the weight matrix of the candidate hidden state and $b_h$ is the bias of the candidate hidden state.

The update gate $r_t$ is calculated using Equation (8) and $\sigma$ is a Sigmoid function which is used as an activation function. Thus, the result of the reset gate is a vector with a value between 0 and 1, which is then used to determine the element of the previous hidden state that should be updated by the current candidate hidden state. For this reason, it is called the update gate.

$$z_t = \sigma(x_t w_{xz} + h_{t-1} w_{hz} + b_z) \tag{8}$$

In this equation, $x_t$ is the current input vector, $h_{t-1}$ is the hidden state of the previous time point, $w_{xz}$ and $w_{hz}$ make up the weight matrix of the update gate, and $b_z$ is the bias of the update gate.

The hidden state is the output of the GRU calculated using the candidate hidden state and the previous hidden state in Equation (9):

$$h_t = z_t \odot h_{t-1} + (1 - z_t) \odot \widetilde{h}_t \tag{9}$$

Here, $z_t$ is the update gate, $h_{t-1}$ is the previous hidden state, $\widetilde{h}_t$ is the current candidate hidden state, and $h_t$ is the current hidden state, which is the ouput of the GRU.

For the GRU, the output is only affected by the current input and the previous hidden state, and is not related to the subsequent state, making the GRU a unidirectional model. However, in order to extract effective features it is necessary to pay attention to the current information, the previous information, and the the subsequent information. It can be easily understood that combining these contexts makes it easier to understand textual information. Based on this idea, BiGRU is proposed to combine forward GRU and backward GRU, making the output result the combination of the two GRUs by weight matrix. The structure of BiGRU is presented in Figure 7; it is composed of an input layer, forward GRU, backward GRU, and output layer. In the figure, $\overrightarrow{h}_t$ is the hidden state of the forward GRU, $\overleftarrow{h}_t$ is the hidden state of the backward GRU, $x_t$ represents the input vectors of BiGRU, $h_t$ is the ouput hidden state of BiGRU, and $x_t \in R^{k \times d}$, where $k$ is the batch size and $d$ is the length of the input vectors. The forward hidden state $\overrightarrow{h}_t \in R^{k \times d}$ and the backward hidden state $\overleftarrow{h}_t \in R^{k \times d}$ are calculated using Equations (10) and (11).

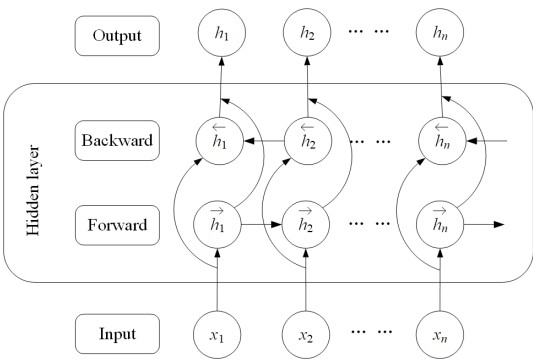

**Figure 7.** Structural diagram of the bidirectional gated recurrent unit [23].

$$\overrightarrow{h_t} = GRU(x_t, \overrightarrow{h_{t-1}}) \tag{10}$$

$$\overleftarrow{h_t} = GRU(x_t, \overleftarrow{h_{t+1}}) \tag{11}$$

In the above equations, *GRU* means the calculation process of the unidirectional GRU, $x_t$ is the input vector, $\overrightarrow{h_{t-1}}$ is the hidden state of the forward GRU, and $\overleftarrow{h_{t+1}}$ is the hidden state of the backward GRU.

The final output hidden state of the BiGRU combines the outputs of the forward and backward GRUs using a weight matrix calculated by Equation (12):

$$h_t = w_{\overrightarrow{h_t}} \overrightarrow{h_t} + w_{\overleftarrow{h_t}} \overleftarrow{h_t} + b_t \tag{12}$$

In the equation, $w_{\overrightarrow{h_t}}$ and $w_{\overleftarrow{h_t}}$ make up the weight matrix of the output layer and $b_t$ is the bias of the output layer.

## 3. Results

The XLNet-BiGRU-Att model was compared with the BERT-BiLSTM, BERT, and XLNet emotional analysis models. The parameter settings of each model are shown in Table 1. In the table, Batch Size is the amount of data for each parameter update, Epochs is the number of iterations in the training process, Embedding is the word vector dimension, and Lr is the learning rate set by the model. The parameters for XLNet-BiGRU-Att proposed in this paper are provided at the bottom of the table.

**Table 1.** Values of model parameters.

| Model | Batch Size | Epochs | Embedding | Lr |
|---|---|---|---|---|
| BERT-BiLSTM | 128 | 50 | 100 | 0.001 |
| XLNet | 256 | 50 | 100 | 0.001 |
| Bert | 64 | 300 | 768 | 0.001 |
| XLNet-BIGRU-Att | 256 | 300 | 768 | 0.001 |

Batch size is an important parameter for the training process, for which sizes of 2, 4, 8, 16, 32, 64, 128, 256, 512..., are typically used. Within the appropriate range, the larger the batch is, the more accurate the descent direction is. Smaller batch sizes cause larger vibration, while larger batch sizes reduce the convergence speed, meaning that more epochs are needed, and may cause memory overflow. Convergence experiments with different batch sizes of 64, 128, and 256 were conducted; the learning curve is shown in Figure 8. It can be seen that the learning curve has the highest vibration with a batch size of 64. Although it reaches the minimum loss very fast, the loss begin to increase after 100 epochs. Therefore the training process with a batch size of 64 is not stable over 400 epochs. When the batch size is 128, the loss decreases slowly, and the model does not reach the best state

after 400 epochs. When the batch size is 256, the learning curve reaches convergence in about 250 epochs, and the vibration is smaller than with batch sizes of 64 or 128. With a batch size of 512 the training process would cause memory overflow. Thus, the best parameters are a batch size of 256 and 300 epochs.

Increasing the dimension of embedding can enhance the richness and meticulousness of text representation to an extent. The embedding dimension of BERT has two versions, 768 and 1024. In order to ensure a fair comparison with BERT, the same embedding dimensions were selected for XLNet-BiGRU-Att. As the 1024 version has three times the parameters of the 768 version, the latter was selected in order to reduce the computational burden.

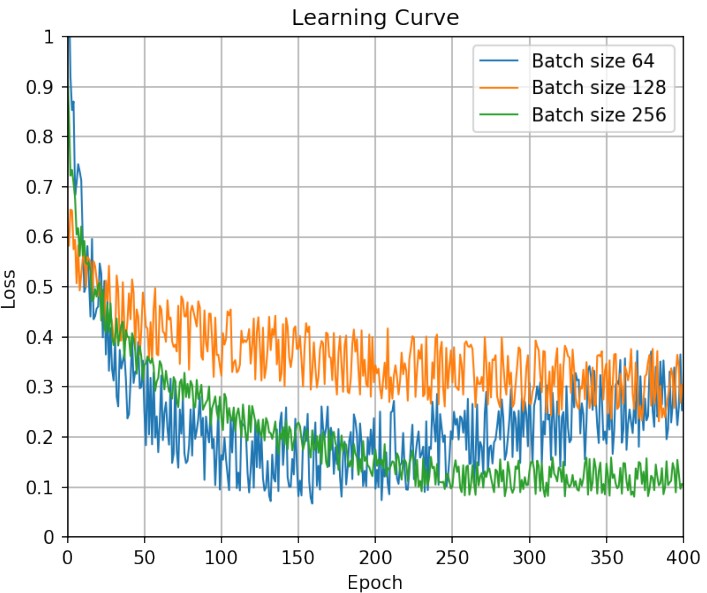

**Figure 8.** Learning curves with different batch sizes.

BERT and XLNet are the basic models which are usually used for text representation, and their results can be directly used for emotion classification with a fully connected network and softmax function. Another approach is to add feature extraction models on top of BERT or XLNet for classification, such as BERT-BiLSTM and the XLNet-BiGRU-Att model proposed in this paper. Therefore, the experimental performance of BERT, XLNet, BERT-BiLSTM, and XLNet-BiGRU-Att were compared on the two datasets introduced in Section 2.1. The experimental results on IEMOCAP are shown in Table 2, and a bar chart is used for visual comparison in Figure 9.

**Table 2.** Evaluation results of different models on the IEMOCAP dataset.

|                 | Accuracy | Precision | Recall | F1-Measure |
|-----------------|----------|-----------|--------|------------|
| BERT-BiLSTM     | 87.92    | 87.25     | 86.26  | 86.75      |
| XLNet           | 88.64    | 88.35     | 87.52  | 87.93      |
| Bert            | 85.86    | 86.11     | 85.64  | 85.87      |
| XLNet-BIGRU-Att | 91.71    | 89.23     | 89.24  | 89.23      |

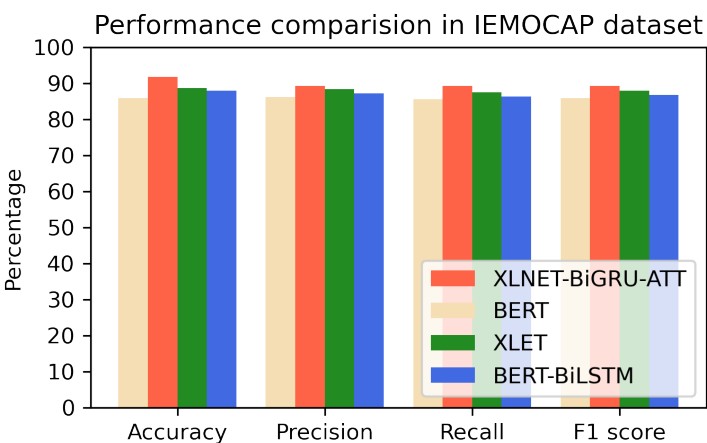

**Figure 9.** Performance comparison on the IEMOCAP dataset.

The same experiments were carried out on the CASIA dataset. The detailed results are shown in Table 3, and a bar chart is shown in Figure 10.

**Table 3.** Evaluation results of different models on the CASIA dataset.

|  | Accuracy | Precision | Recall | F1-Measure |
| --- | --- | --- | --- | --- |
| BERT-BiLSTM | 81.92 | 79.91 | 83.64 | 81.73 |
| XLNet | 83.64 | 79.75 | 82.41 | 81.05 |
| Bert | 79.86 | 76.27 | 80.43 | 78.29 |
| XLNet-BIGRU-Att | 85.71 | 80.60 | 84.60 | 82.55 |

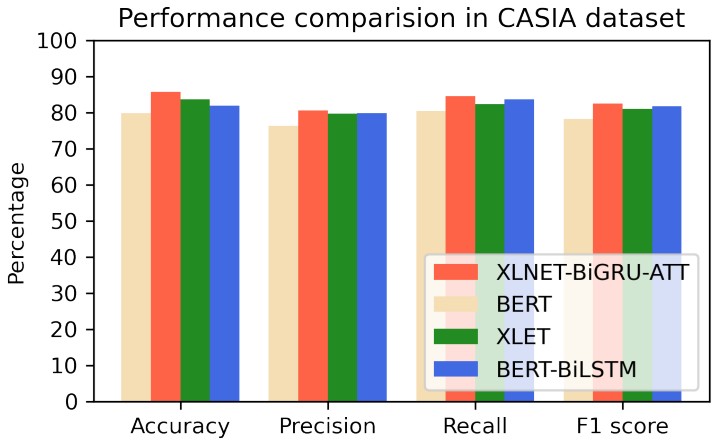

**Figure 10.** Bar chart showing the evaluation results of different models on the CASIA dataset.

From the experimental results, it can be seen that the pre-trained models can calculate effective word vectors for TER, and they all perform well on the both datasetsto calculate the word vector. However, compared to the others, the model proposed in this paper has the best accuracy on the IEMOCAP and CASIA datasets, reaching 91.17% and 85.71%, respectively.

## 4. Discussion

Upon examining the respective datasets, it is apparent that the IEMOCAP dataset provides superior performance results in comparison to the CASIA dataset. IEMOCAP contains nine distinct emotions: Neutral, Happy, Sad, Angry, Surprise, Fear, Disgust, Excited, and Other. Nonetheless, the distribution of sample categories in the nine-emotion dataset is noticeably imbalanced, with the majority of samples belonging to the four

fundamental emotions (Happy, Sad, Angry, and Neural). As a consequence, many academic studies concentrate solely on these basic emotions. To enable a more accurate comparison of the results, this paper similarly focuses on the four basic emotions within the IEMOCAP dataset. On the contrary, the CASIA dataset comprises six emotions, thereby posing a greater challenge for model classification. Hence, based on the inherent difficulty of the classification problem, it is established that the IEMOCAP dataset yields more favorable results than the CASIA dataset.

The experiment primarily focused on a comparison of two models, BERT and XLNet, with regard to text emotion recognition. The absolute values of the results indicate that both models are proficient in this area. However, the relative results suggest that XLNet outperforms BERT due to its employment of attention, larger hidden layers, and additional positional encoding elements. On the other hand, BERT-BiLSTM primarily prioritizes optimization of the bidirectional LSTM network structure, attention mechanism, and long-term dependency modeling, resulting in better performance than BERT on related tasks. Our experiments indicate that the performance of BERT-BiLSTM is comparable to that of XLNet.

Although both BERT and XLNet perform well on tasks involving semantic representation of textual information, our experiments show that their performance when directly applying semantic representation results to emotion classification is not as good as that attained by models with additional feature extraction capability. For example, BERT-BiLSTM performs better than BERT on both datasets; similarly, XLNet-BILSTM-Att performs better than XLNet. Therefore, the above experiments demonstrate that the semantic representations of BERT and XLNet involve general natural language understanding. When dealing with specialized tasks such as emotion recognition, adding specific feature extraction modules can improve the performance of models.

This paper presents enhancements to XLNet achieved by incorporating bidirectional gate recurrent units and an attention masking mechanism into its architecture. The bidirectional transmission feature of BiGRU serves to effectively process long sequence data, making the model better suited for text mining applications. Additionally, the attention mask mechanism connects contextual semantics and maintains the dependency of multiple word predictions, which improves overall model accuracy. The performance of XLNet is boosted when supported by bidirectional gate recurrent units and attention masking mechanisms, as shown by our experimental results. Specifically, the proposed XLNet-BiGRU-Att model achieved the highest accuracy of all tested models, reaching at 91.71% on the IEMOCAP dataset and 85.71% on the CASIA dataset.

## 5. Conclusions

Text emotion recognition is a crucial aspect of natural language understanding that finds extensive application in domains such as human–computer interaction, public opinion analysis, mental health analysis, and social network analysis. The development of advanced technologies in the fields of computing, the internet, and artificial intelligence has considerably amplified the significance of this field. In this paper, the XLNet-BiGRU-Att model is proposed, building on XLNet by leveraging BiGRU and Att to achieve enhanced performance. This paper has presented and analyzed the structure of the model and described the experiments we carried out on the IEMOCAP and CASIA datasets. The results show that our proposed XLNet-BiGRU-Att model outperforms BERT, BERT-BiLSTM, and XLNet, achieving an accuracy of 91.71% on IEMOCAP and 85.71% on CASIA. The theoretical and experimental analysis and discussion presented in this paper confirm the suitability of XLNet-BiGRU-Att for text emotion recognition.

**Author Contributions:** Methodology, T.H.; Writing—original draft preparation, Z.Z.; Conceptualization, C.D.; Data curation, M.R.; Supervision, X.J.; Software, Q.Z.; Project administration, T.H. All authors have read and agreed to the published version of the manuscript.

**Funding:** This research was funded by the Jinhua Science and Technology Bureau, grant number 2022-1-046, and by the Jinhua Advanced Research Institute, grant number G202209 and G202207.

**Institutional Review Board Statement:** Not applicable.

**Informed Consent Statement:** Informed consent was obtained from all subjects involved in the study.

**Data Availability Statement:** Publicly available datasets were analyzed in this study. The data can be found at: http://www.chineseldc.org/resource_info.php?rid=76 and https://sail.usc.edu/iemocap/ iemocap_release.htm (accessed on 1 January 2023).

**Acknowledgments:** This research was supported by the Jinhua Science and Technology Bureau and the Jinhua Advanced Research Institute.

**Conflicts of Interest:** The authors declare no conflict of interest.

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
