# Peer review of "Text Emotion Recognition Based on XLNet-BiGRU-Att"

_electronics, doi:10.3390/electronics12122704_

Round 1

Reviewer 1 Report

The paper presents an analysis of XLNet- 334 BiGRU-Att model and evaluates its efficacy through experiments on two emotional text datasets - IEMOCAP and CASIA. The paper is well structured and proposes interesting results. I would recommend to add a description of the goal of the study as well as research questions that were posed for this study. Also, a comparison with the previous papers in the Discussion section could enhance a scientific soundness.

Reviewer 2 Report

The current version of the paper is improved.

However, all acronyms must be defined including IEMOCAP and CASIA in the abstract.

Variables in the Fig. 1 must be defined.

More comments about the hyperparameters tuning must be provided (Table 1 - Value of the model parameter).

English language is okay.

Reviewer 3 Report

I think the subject of the manuscript is worthy of investigation and suitable for the journal. The authors propose a novel approach addressing the challenges associated with vectorization of short text sentiment analysis through the development of a model based on XLNet with bidirectional recurrent unit and attention mechanism (XLNet-BiGRU-Att).

The incorporation of bidirectional gate recurrent units and attention masking mechanisms into proposed architecture allows to effectively process long sequence data in text mining applications.

The authors integrate three well-known methods to increase the accuracy, precision and recall of the determination of the emotional coloring of the text. But I would like to see a single example of the use of the method, from which you can make a clear idea of ​​how the specified methods are integrated. I have in mind a small text and the step by step determination of a text emotion by means of the proposed method.

Revealed mistakes do not allow a presentation of the manuscript in its current form. In this regard, I recommend that the authors revise the manuscript in response to the comments and resubmit it.

Common comments

One-hot encoding is not a method, it is a process or technique.

There are some issues in Section 1. Paper 7 is not about One-hot encoding. Paper 13 is not about BERT. Paper 16 does not propose XLNet. Paper 17 does not propose RoBERTa and DistilBERT models.

The variables in figures 1, 2, and 3 should be described on the same page/figure as well as arrows and blocks.

Line 106, what is the AR model?

Line 129, what is the sequence of D? The variable D is not defined.

The authors should fix the figure 3 by indicating the direction of information flow.

What is the difference between time point t and time step h?

What is the sense of the variable d? What is the number of input moments?

The elements of figure 5 should be described.

What does the T sign mean in equations 10 and 11?

What does the number in parentheses mean in Figure 6?

Figure 6 does not contain parts (a) and (b).

Line 295, where is the mentioned algorithm?

English still needs improvements.

Minor editing of English language required

Reviewer 4 Report

This paper proposes a veriation of the XLNet deep learning model with the addition of a BiGRu and self attention layers. The model,  from the results reported, outperforms other competing models although the models used for the comparision are, perhaps, not the latest. I would have considered RoBERTa and DistilBERT. Also, in the past year or so the trend is to move away from RNNs, LSTMs, and GRUs and to build deep learning models using attention only. 

The paper is well written and the standard of English is high. In fact, it is quite good and there are only a few typos that are listed below.

I think that this work is of interest to the community working in text emotion recognition. The proposed model does perform better on the chosen datasets. Havign said that it would be interesting to see how the model would perform in comparision to attention-only models. These models can attend to much longer sequences although one might argue that emotion tends to occur in a local context only.

Section 2.3 needs some reworking to make clearer.

List of typos and comments:

Line 106 - XLNET is based on the AR language model.

Line 114 - an example of prediction based on the

Equation 1 - what is theta? Is lg the natural log (ln) or log to base 2?

Line 238 - Equation (10)

Line 283 - the superiority of the proposed model is demonstrated.

Line 287 - used for visual comparision.

The quality of English is good and commensurate with that expected in such a journal. A number of typos are listed above.
